# Virtual-photon-mediated spin-qubit–transmon coupling

A.J. Landig[1]*, J.V. Koski[1], P. Scarlino[1], C. Müller[2], J.C. Abadillo-Uriel[3], B. Kratochwil[1], C. Reichl[1], W. Wegscheider[1], S.N. Coppersmith [3,4], Mark Friesen [3], A. Wallraff [1], T. Ihn[1] & K. Ensslin [1]

Spin qubits and superconducting qubits are among the promising candidates for realizing a solid state quantum computer. For the implementation of a hybrid architecture which can profit from the advantages of either approach, a coherent link is necessary that integrates and controllably couples both qubit types on the same chip over a distance that is several orders of magnitude longer than the physical size of the spin qubit. We realize such a link with a frequency-tunable high impedance SQUID array resonator. The spin qubit is a resonant exchange qubit hosted in a GaAs triple quantum dot. It can be operated at zero magnetic field, allowing it to coexist with superconducting qubits on the same chip. We spectroscopically observe coherent interaction between the resonant exchange qubit and a transmon qubit in both resonant and dispersive regimes, where the interaction is mediated either by real or virtual resonator photons.

[1] Department of Physics, ETH Zürich, CH-8093 Zürich, Switzerland. [2] IBM Research Zurich, CH-8803 Rüschlikon, Switzerland. [3] Department of Physics, University of Wisconsin-Madison, Madison, WI 53706, USA. [4] Present address: School of Physics, University of New South Wales, Sydney, NSW 2052, Australia. *email: alandig@phys.ethz.ch

A future quantum processor will benefit from the advantages of different qubit implementations[1]. Two prominent workhorses of solid state qubit implementations are spin- and superconducting qubits. While spin qubits have a high anharmonicity, a small footprint[2] and promise long coherence times[3–5], superconducting qubits allow fast and high fidelity read-out and control[6,7]. A coherent link, which couples both qubit systems controllably over distances exceeding the physical size of the spin qubit, typically hundreds of nanometers, by several orders of magnitude is required to create an integrated scalable quantum device. An architecture to provide such a link is circuit quantum electrodynamics (circuit QED)[8], where microwave photons confined in a superconducting resonator couple coherently to the qubits. Circuit QED was initially developed for superconducting qubits[9], where long-distance coupling[10,11] enables two-qubit gate operations[12]. Recently, coherent qubit-photon coupling was demonstrated for spin qubits[13–15] in few electron quantum dots. However, coupling a spin qubit to another distant qubit[16,17] has not yet been shown. One major challenge for an interface between spin and superconducting qubits[18] is that spin qubits typically require large magnetic fields[19,20], to which superconductors are not resilient[21].

We overcome this challenge by using a spin qubit which relies on exchange interaction[22]. This resonant exchange (RX) qubit[23–27] is formed by three electrons in a GaAs triple quantum dot (TQD). We implement the qubit at zero magnetic field without reducing its coherence compared to earlier measurements at finite magnetic field[15]. The quantum link is realized with a frequency-tunable high impedance SQUID array resonator[28], which couples the RX and the superconducting qubit coherently over a distance of a few hundred micrometers. The RX qubit coupling strength to the resonator and its decoherence rate are tunable electrically. We find that their ratio is comparable to previously reported values for spin qubits in Si[13,14]. We demonstrate coherent coupling between the two qubits first by resonant and then by virtual photon exchange in the high impedance resonator. We electrostatically tune the RX qubit to different regimes, where the qubit states have either a dominant spin or charge character.

## Results

**Sample and qubit characterization.** The design of our sample is illustrated schematically in Fig. 1a. It is similar to ref. [29], where a semiconductor charge qubit was used instead of a spin qubit. An optical micrograph can be found in the Methods section. The superconducting qubit we use is a transmon[30,31] as its Josephson energy exceeds the charging energy by about two orders of magnitude (see characterization below). The transmon consists of an Al SQUID grounded on one side and connected in parallel to a shunt capacitor. We tune the transition frequency $\nu_T$ between the transmon ground $|0_T\rangle$ and first excited state $|1_T\rangle$ by changing the flux $\Phi_T$ through the SQUID loop with an on-chip flux line.

The transmon and the RX qubit are capacitively coupled to the same end of a SQUID array resonator, which we denote as coupling resonator in the following, with electric dipole coupling strengths $g_T$ and $g_{RX}$. The other end of the coupling resonator is connected to DC ground. It is fabricated as an array of Al SQUID loops[28], which enables the tuning of its resonance frequency $\nu_C$ from $\sim 4 - 7$ GHz within the detection bandwidth of our measurement setup with a magnetic flux $\Phi_C$ produced by a coil mounted close to the sample. In addition, the resonator has a high characteristic impedance that enhances its coupling strength to both qubits (see Supplementary Note 1). The transmon flux $\Phi_T$ has a negligible effect on $\nu_C$.

The transmon is also capacitively coupled to a 50 $\Omega \lambda/2$ coplanar waveguide resonator with a coupling strength $g_R/2\pi \simeq 141$ MHz. Throughout this article, we refer to this resonator as the read-out resonator, because it allows us to independently probe the transmon without populating the coupling resonator with photons. The read-out resonator has a bare resonance frequency $\nu_R = 5.62$ GHz and a total photon decay rate $\kappa_R/2\pi = 5.3$ MHz. As illustrated in Fig. 1a, the coupling and the read-out resonator are probed by measuring the reflection of a multiplexed probe tone at frequency $\nu_p$. In addition, we can apply a drive tone at frequency $\nu_d$ that couples to both qubits via the resonators. For the experiments presented in this work, the probe tone power is kept sufficiently low for the estimated average number of photons in both resonators to be less than one.

In Fig. 1b we characterize the transmon with two-tone spectroscopy. The first tone probes the read-out resonator on resonance ($\nu_p = \nu_R$), while the second tone is a drive at frequency $\nu_d$ that is swept to probe the transmon resonance. Once $\nu_d = \nu_T$, the transmon is driven to a mixed state, which is observed as a change in the resonance frequency of the dispersively coupled read-out resonator. This frequency shift is detected with a standard heterodyne detection scheme[32] as a change in the complex amplitude $A = I + iQ$ of the signal reflected from the resonator. In Fig. 1b, centered at $\nu_d = \nu_T(\Phi_T)$ we observe a peak in $|A - A_0|$. Here, $A_0$ is the complex amplitude in the absence of the second (drive) tone. From a fit of the transmon dispersion to the multi-level Jaynes-Cummings model and by including the position of higher excited states of the transmon probed by two photon transitions (not shown)[33,34], we obtain the maximum Josephson energy $E_{J,max} = 18.09$ GHz and the transmon charging energy $E_c = 0.22$ GHz (for details see Supplementary Note 2). Note that the parameters for all theory fits in this article can be found in Supplementary Note 4.

At a distance of a few hundred micrometers from the transmon SQUID, we form a TQD by locally depleting a two-dimensional electron gas in a GaAs/AlGaAs heterostructure with the Al top gate electrodes shown in Fig. 1c. One of the electrodes extends to the coupling resonator to enable electric dipole interaction between photons and TQD states. Another electrode allows us to apply RF signals at frequency $\nu_{dRX}$. We use a QPC charge detector to help tune the TQD to the three electron regime. The symmetric $(1,1,1)$ and the asymmetric $(2,0,1)$ and $(1,0,2)$ charge configurations are relevant for the RX qubit as they are lowest in energy. We sweep combinations of voltages on the TQD gate electrodes to set the energy of the asymmetric configurations equal and control the energy detuning $\Delta$ of the symmetric configuration with respect to the asymmetric ones (see Fig. 1d).

There are two spin states within $(1,1,1)$ that have $S = S_z = 1/2$ equal to the spin of two states with asymmetric charge configuration, which form a singlet in the doubly occupied dot. An equivalent set of states with $S = 1/2, S_z = -1/2$ exists. As the resonator response is identical for both sets of states, considering only one is sufficient (see Supplementary Note 3 for a detailed discussion). This results in a total of four relevant states for the qubit[15]. The tunnel coupling $t_l$ ($t_r$) between the left (right) quantum dot and the middle quantum dot hybridizes these states, which leads to the formation of the two RX qubit states $|0_{RX}\rangle$ and $|1_{RX}\rangle$. For $\Delta < 0$, $|0_{RX}\rangle$ and $|1_{RX}\rangle$ have predominantly the $(1,1,1)$ charge configuration but different spin arrangement. Consequently, with increasingly negative $\Delta$, the spin character of the qubit increases, which reduces the qubit dephasing due to charge noise. This comes at the cost of a reduced admixture of asymmetric charge states and therefore a decrease in the electric dipole coupling strength $g_{RX}$. In contrast, for $\Delta > 0$ the RX qubit states have dominantly the asymmetric charge configurations $(2,0,1)$ and $(1,0,2)$. The qubit therefore has a dominant charge

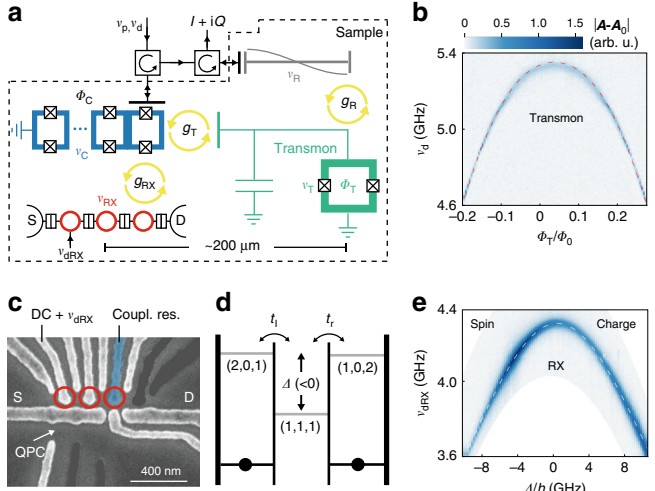

**Fig. 1** Sample and qubit dispersions. **a** Schematic of sample and measurement scheme. The signals at frequencies $\nu_p$ (probe) and $\nu_d$ (drive) are routed with circulators as indicated by arrows. The reflected signal $I + iQ$ at $\nu_p$ is measured. The sample (dashed line) contains four quantum systems with transition frequencies $\nu_i$: a coupling resonator that consists of an array of SQUID loops ($\nu_C$, blue), an RX qubit ($\nu_{RX}$, red), a transmon ($\nu_T$, green) and a read-out resonator ($\nu_R$, gray). Empty black double-rectangles indicate electron tunnel barriers separating the three quantum dots (red circles) as well as the source (S) and drain (D) electron reservoirs. A drive tone at frequency $\nu_{dRX}$ can be applied to one of the dots. Crossed squares denote the Josephson junctions of SQUIDs. Yellow arrows indicate the coupling between the quantum systems with coupling strengths $g_i$. $\Phi_C$ and $\Phi_T$ denote coupling resonator and transmon flux, respectively. **b** Two-tone spectroscopy of the transmon, with the RX qubit energetically far detuned. We plot the complex amplitude change $|A - A_0|$ (see main text) as a function of drive frequency $\nu_d$ and $\Phi_T/\Phi_0$. The dashed line indicates $\nu_T$ as obtained from the system Hamiltonian. **c** Scanning electron micrograph of the TQD and quantum point contact (QPC) region of the sample. Unused gate lines are grayed out. The gate line extending to the coupling resonator is highlighted in blue. **d** TQD energy level diagram indicating the tunnel couplings $t_l$ and $t_r$ and the electrochemical potentials, parametrized by $\Delta$, of the relevant RX qubit states $(N_l, N_m, N_r)$ with $N_l$ electrons in the left, $N_m$ electrons in the middle and $N_r$ electrons in the right quantum dot. **e** Two-tone spectroscopy of the RX qubit, with the transmon energetically far detuned for $\nu_p \simeq \nu_C = 4.84$ GHz as a function of $\Delta$ and $\nu_{dRX}$. The dashed line shows the expected qubit energy obtained from the Hamiltonian of the system

character, which increases, together with $g_{RX}$, with increasing positive $\Delta$. Independent of $\Delta$, the RX qubit states have the same total spin and spin $z$-component such that they can be driven directly by electric fields[35] and be operated in the absence of an applied external magnetic field. This is in contrast to other spin qubit implementations, which rely on engineered or intrinsic spin-orbit interaction[36–41] for spin-charge coupling.

Four similar RX qubit tunnel coupling configurations were used in this work as listed in the Methods section. We use two-tone spectroscopy[33] to characterize the RX qubit dispersion: we apply a probe tone on resonance with the coupling resonator, drive the qubit via the gate line and tune its energy with $\Delta$. The spectroscopic signal in Fig. 1e agrees with the theoretically expected qubit dispersion for qubit tunnel coupling configuration 3 (see the Methods section).

**Resonant interaction**. First, we investigate the resonant interaction between the coupling resonator and the RX qubit. To start with, both qubits are detuned from the coupling resonator. Then, we sweep $\Delta$ to cross a resonance between the RX qubit and the resonator, while keeping the transmon far detuned. We observe a well resolved avoided crossing in the $|S_{11}|$ reflectance spectrum shown in Fig. 2a and extract a spin qubit-photon coupling strength of $g_{RX}/2\pi = 52$ MHz from a fit to the vacuum Rabi mode splitting shown in black in Fig. 2c. This coupling strength enhancement compared to earlier work in ref. [15] is an important ingredient to realize coherent virtual-photon-mediated qubit–qubit interaction, which is weaker compared to resonant qubit-photon interaction. The enhancement is related to an enhanced characteristic impedance of the resonator, to the position of the qubit at the open end of the resonator, where the voltage vacuum fluctuations are maximal, as well as to an

optimized TQD gate design with an increased overlap of the resonator gate with the underlying quantum dot. The spin qubit and the coupling resonator photons are strongly coupled since $g_{RX} > \kappa_C, \gamma_{2,RX}$, with the RX qubit decoherence rate $\gamma_{2,RX}/2\pi = 11$ MHz and the bare coupling resonator linewidth $\kappa_C/2\pi = 4.6$ MHz. The decoherence rate is determined independently with power dependent two-tone spectroscopy. We dispersively detune the coupling resonator with $\Phi_C$ from the RX qubit and extrapolate the width of the peak observed in the two-tone spectroscopy response (c.f. Fig. 1e) to zero drive power[33].

Next, we characterize the interaction between the transmon and the coupling resonator. We tune the transmon through the resonator resonance by sweeping $\Phi_T$. For this measurement, the RX qubit is far detuned in energy. We resolve the hybridized states of the transmon and the resonator photons in the measured $|S_{11}|$ spectrum in Fig. 2b. They are separated in energy by the vacuum Rabi mode splitting $2g_T/2\pi = 360$ MHz illustrated in Fig. 2c in green. We perform power dependent two-tone spectroscopy to extract the transmon linewidth by probing the read-out resonator. We obtain $\gamma_{2,T}/2\pi = 0.7$ MHz, which we estimate to be limited by Purcell decay[42,43]. Consequently, the strong coupling limit $g_T > \kappa_C, \gamma_{2,T}$ is also realized for transmon and coupling resonator.

We now demonstrate that the two qubits interact coherently via resonant interaction with the coupling resonator. For this purpose, we first tune the transmon and the coupling resonator into resonance, where the hybrid system forms the superposition states $|\pm\rangle = (|0_T, 1_C\rangle \pm |1_T, 0_C\rangle)/\sqrt{2}$ of a single excitation in either the resonator or the qubit. Then, we sweep $\Delta$ to tune the RX qubit through a resonance with both the lower energy state $|-\rangle$ and the higher energy state $|+\rangle$. In the $|S_{11}|$ spectrum in

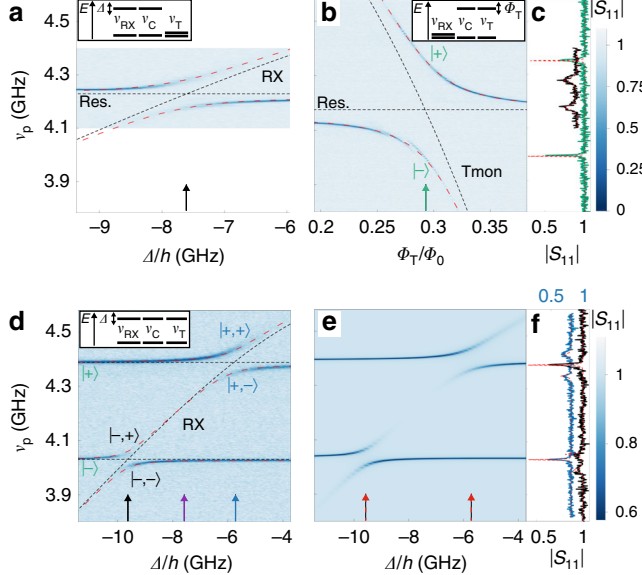

**Fig. 2** Resonant interaction. The schematics at the top of the graphs indicate the energy levels of the RX qubit ($\nu_{RX}$), coupling resonator ($\nu_C$) and transmon ($\nu_T$). Theory curves in the absence (presence) of coupling are shown as dashed black (red) lines. **a** Reflected amplitude $|S_{11}|$ as a function of RX detuning $\Delta$ and probe frequency $\nu_p$ for RX qubit tunnel coupling configuration 2. **b** Reflected amplitude $|S_{11}|$ as a function of relative transmon (Tmon) flux $\Phi_T/\Phi_0$ and $\nu_p$. The states $|\pm\rangle$ are discussed in the main text. **c** Cuts from panel **a** at $\Delta/h \simeq -7.6$ GHz (black) and from panel **b** at $\Phi_T/\Phi_0 \simeq 0.3$ (green) as marked with arrows in the respective panels. The black trace is offset in $|S_{11}|$ by 0.1. Theory fits are shown as red dashed lines. **d** $|S_{11}|$ as a function of $\Delta$ and $\nu_p$ for RX qubit tunnel coupling configuration 2. The states $|-, \pm\rangle$ and $|+, \pm\rangle$ are explained in the main text. The black and blue arrows are referred to in **f**, the purple arrow is discussed in the text. **e** Result of master equation simulation for parameters as in **d**. The values for $|S_{11}|$ are scaled to the experimental data range in **d**. **f** $|S_{11}(\nu_p)|$ at $\Delta/h \simeq -9.8$ GHz and $\Delta/h \simeq -5.6$ GHz as marked with the corresponding colored arrows in panels **d**, **e**. The blue trace is offset in $|S_{11}|$ by 0.2

Fig. 2d, avoided crossings are visible at both resonance points. This indicates the coherent interaction of the three quantum systems which form the states $|-, \pm\rangle$ and $|+, \pm\rangle$, where the second label indicates a symmetric or antisymmetric superposition of the RX qubit state with the transmon-resonator $|\pm\rangle$ states. The splitting $2g_\mp$ between $|-, \pm\rangle$ and $|+, \pm\rangle$ is extracted from the $|S_{11}|$ reflection measurements in Fig. 2f. We obtain $2g_+/2\pi = 84$ MHz at $\Delta/h \simeq -5.6$ GHz and $2g_-/2\pi = 63$ MHz at $\Delta/h \simeq -9.8$ GHz from the fits in Fig. 2f. The smaller $g_-$ compared to $g_+$ is due the decrease of the RX qubit dipole moment with more negative $\Delta$. The RX qubit, the transmon and the resonator are on resonance ($\nu_{RX} = \nu_T = \nu_C$) between the avoided crossings in Fig. 2d at $\Delta/h \simeq -7.8$ GHz (see purple arrow). There, the splitting of the dips in the reflection spectrum is enhanced by $\approx 16$ MHz compared to the off-resonant splitting of $|\pm\rangle$ at $\Delta/h \approx -11.4$ GHz in Fig. 2d (see Supplementary Note 1). This enhancement is an experimental signature of the coherent resonant interaction of all three quantum systems in good agreement with the theoretical value $(2g_T - 2\sqrt{g_T^2 + g_{RX}^2})/2\pi \approx 15$ MHz calculated from independently extracted parameters.

The experimental observation in Fig. 2d is well reproduced by a quantum master equation simulation shown in Fig. 2e and further discussed in Supplementary Note 2.

**RX qubit optimal working point**. While $\gamma_{2,T}$ is limited by Purcell decay and therefore does not depend on $\Phi_T$, $\gamma_{2,RX}$ changes with $\Delta$[15]. For obtaining the data shown in Fig. 3a we use power dependent two-tone spectroscopy via the coupling resonator to measure $\gamma_{2,RX}$ as a function of $\Delta$. We observe an increase of $\gamma_{2,RX}$ as the charge character of the qubit is increased with $\Delta$. Compared to ref. [15], the data in Fig. 3a covers a larger range in $\Delta$, in particular for $|\Delta| \gg t_{l,r}$. The data suggests a lower limit of $\gamma_{2,RX}/2\pi \simeq 6.5$ MHz for $\Delta \ll 0$. This is in agreement with refs. [44] and [15], where the RX qubit was operated at a finite magnetic field of a few hundred mT. Hence, our experiment indicates that the RX qubit can be operated near zero magnetic field without reducing its optimal coherence. In our experiment, the maximum external magnetic field determined by $\Phi_C$ is of the order of 1 mT. To model the RX qubit decoherence in Fig. 3a, we consider an ohmic spectral density for the charge noise as well as the hyperfine field of the qubit host material that acts on the spin part of the qubit (see Supplementary Notes 2 and 3). Theory and experiment in Fig. 3a match for a width $\sigma_B = 3.48$ mT of the hyperfine fluctuations in agreement with other work on spin in GaAs[45–47]. This suggests that $\gamma_{2,RX}$ is limited by hyperfine interactions.

The colored data points in Fig. 3a were measured for a smaller RX qubit-coupling resonator detuning compared to the black data points (numbers are given in Fig. 3 caption). The smaller detuning is used for the virtual interaction measurements discussed below. We observe an increase of $\gamma_{2,RX}$ for small qubit-resonator detuning compared to large detuning. This increase is about one order of magnitude larger than our estimated difference of Purcell decay and measurement induced dephasing for those different data sets (see Supplementary Note 4). In contrast, for the transmon that is insensitive to charge noise, we do not observe this effect. This suggests that the effect is due to charge noise induced by the coupling resonator.

As $\gamma_{2,RX}$ increases with $\Delta$ in Fig. 3a, the RX qubit coupling strength $g_{RX}$ to the coupling resonator increases. This implies the existence of an optimal working point for the RX qubit, where $g_{RX}/\gamma_{2,RX}$ is maximal. While a distinct optimal point is not discernible for the black data points in Fig. 3b, the averaged value of $g_{RX}/\gamma_{2,RX} \simeq 9$ in the spin dominated regime for $-6 < \Delta/h < 0$ GHz is about a factor of 1.7 larger than values reported so far for Si spin qubits[13,14]. In contrast, for the colored data points we observe an optimal working point at $\Delta/h \simeq -3.3$ GHz since $g_{RX}/\gamma_{2,RX}$ is reduced at small qubit-resonator detuning in Fig. 3b compared to the black data points at large detuning due to the influence of the coupling resonator on $\gamma_{2,RX}$ discussed above.

**Virtual photon coupling**. In the following, we investigate the RX qubit-transmon interaction mediated by virtual photons in the coupling resonator at the RX qubit working points marked in color in Fig. 3a. The two qubits are resonant while the coupling resonator is energetically detuned, such that the photon excitation is not dominant in the superposed two-qubit eigenstates. This coupling scheme, illustrated in Fig. 3c, is typically used for superconducting qubits to realize two-qubit operations[12]. We measure the virtual coupling at the optimal working point ($\Delta/h \simeq -3.3$ GHz), at $\Delta/h \simeq -9.9$ GHz and at $\Delta/h \simeq 10.2$ GHz, where $\gamma_{2,RX}$ in Fig. 3a saturates, as well as in the intermediate regime at $\Delta/h \simeq 3.4$ GHz. While the RX qubit is tuned through a resonance with the transmon by changing $\Delta$, they are both detuned by $\Delta_C \equiv \nu_C - \nu_T \simeq 3g_T$ from the coupling resonator. To realize this detuning for every working point, we adjust the qubit and resonator energies with $\Phi_T$, $t_{l,r}$

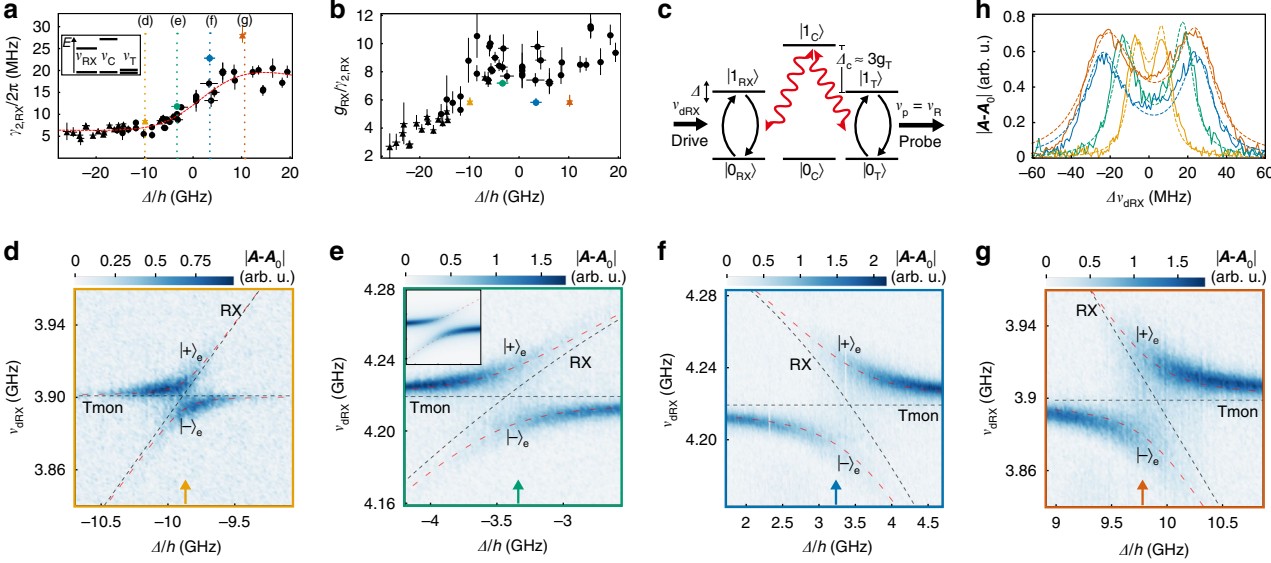

**Fig. 3** RX qubit working points and virtual-photon-mediated interaction. **a** RX qubit decoherence rate $\gamma_{2,\mathrm{RX}}$ as a function of detuning $\Delta$. The dotted vertical lines specify the four working points used in **d–g**. The corresponding colored data points were obtained for a coupling resonator-RX qubit detuning of $\nu_C - \nu_{\mathrm{RX}} \simeq (13.7, 8.0, 5.1, 4.4) g_{\mathrm{RX}}$ for $\Delta/h \simeq (-9.9, -3.3, 3.4, 10.2)$ GHz and the RX qubit tunnel coupling configurations 3 (circle) and 4 (triangle). For the black data points, $\nu_C - \nu_{\mathrm{RX}} \geq 9.7\, g_{\mathrm{RX}}$ with qubit tunnel coupling configuration 1 (circle) and 2 (triangle). The dashed red line is a fit of a model (see main text) to the black data points. Error bars indicate the standard error of fits and an estimated uncertainty of the RX qubit energy of 50 MHz. **b** Ratio of $g_{\mathrm{RX}}$, as obtained from theory, and $\gamma_{2,\mathrm{RX}}$ as shown in **a**. The color and shape code of the data points is the same as in **a**. **c** Schematic of the measurement scheme. Bare qubit transitions (black arrows) are coupled by virtual photon excitations (red arrows) in the detuned coupling resonator ($|0/1\rangle_C$ are the two lowest photon number states). The RX qubit is driven at frequency $\nu_{\mathrm{dRX}}$, the transmon is probed via the read-out resonator at frequency $\nu_R$. **d–g** Two-tone spectroscopy at $\nu_p = \nu_R \simeq 5.6$ GHz as a function of $\Delta$ and drive frequency $\nu_{\mathrm{dRX}}$. Dashed black (red) lines indicate transmon (Tmon) and RX qubit energies in the absence (presence) of coupling. The frame color refers to the RX qubit working points as specified in **a**. The inset in **e** shows the result from a master equation simulation with the same axes as the main graph. **h** Two-tone spectroscopy response from panels **d–g** at $\Delta$ as specified with arrows in the corresponding panels. The cuts are centered around zero by accounting for a frequency offset $\nu_{\mathrm{dRX},0} \equiv \nu_{\mathrm{dRX}} - \Delta\nu_{\mathrm{dRX}}$. The dashed lines show the corresponding theory results

and $\Phi_C$. We drive the RX qubit at frequency $\nu_{\mathrm{dRX}}$ (see Fig. 1a) and investigate its coupling to the transmon by probing the dispersively coupled read-out resonator at its resonance frequency ($\nu_p = \nu_R \simeq 5.6$ GHz). This measurement is shown in Fig. 3d for the working point at $\Delta/h \simeq -9.9$ GHz. For large transmon-spin qubit detuning ($\Delta/h \ll -10$ GHz), the spectroscopic signal of the transmon is barely visible as the drive mainly excites the bare RX qubit. The signal increases with $\Delta$ as the RX qubit approaches resonance with the transmon, such that driving the RX qubit also excites the transmon due to their increasing mutual hybridization. On resonance, we resolve the two hybridized spin-qubit-transmon states $|\pm\rangle_e \simeq (|0_{\mathrm{RX}}, 1_T\rangle \pm |1_{\mathrm{RX}}, 0_T\rangle)/\sqrt{2}$ by about a line width. These states are separated in energy by the virtual-photon-mediated exchange splitting $2J \simeq 2 g_{\mathrm{RX}} g_T / \Delta_C$. The splitting is enhanced at the other working points in Fig. 3e–g, for which the RX qubit control parameter $\Delta$ and consequently $g_{\mathrm{RX}}$ is larger. The result of a master equation simulation shown in Fig. 3e agrees well with the experimental observation. The influence of the RX qubit decoherence rate $\gamma_{2,\mathrm{RX}}$ on the virtual interaction measurement is quantified in Fig. 3h, where we show averaged measurements of the two-tone spectroscopy signal from Fig. 3d–g at $\Delta$ as indicated by arrows in the corresponding panels. The fits of a master equation model in Fig. 3h show excellent quantitative agreement with the experimental curves. As discussed in detail in Supplementary Note 3, fit parameters previously obtained from Fig. 2 were adjusted to account for significant power broadening in these measurements. The exchange splitting is best resolved at the optimal working point, corresponding to

the solid green curve in Fig. 3h, where we obtain $2J/2\pi \simeq 32$ MHz from the fit.

## Discussion

In conclusion, we have implemented a coherent link between an RX qubit and a transmon. The link either utilizes real or virtual microwave photons for the qubit–qubit interaction. The RX qubit was operated in both spin and charge dominated regimes. We found an optimal working point at which the ratio between its resonator coupling and its decoherence rate is maximal and comparable to state of the art values achieved with spin qubits in Si. We also reported that the coupling resonator potentially introduces charge noise that can have significant impact on the RX qubit coherence. The performance of the quantum link in this work is limited by the minimum decoherence rate of the qubit, which is determined by hyperfine interaction in the GaAs host material. Once the spin coherence is enhanced by using hyperfine free material systems such as graphene[48,49] or isotopically purified silicon[50], the spin could be used as a memory that can be coupled on-demand to the transmon by pulsing the qubit control parameter. While three-electron spin qubits have already been implemented in Si[51,52], a sufficiently high transition frequency allowing for circuit QED experiments to be performed, which requires large inter-dot tunnel couplings, has not yet been realized. Compared to GaAs, electrostatic control of single electrons can be more challenging related to overlapping fine gate structures for silicon and graphene quantum dots as well as due to the smaller quantum dot size in silicon. An additional potential challenge in both materials is the valley degree of freedom[35]. As

the coherence properties of the RX qubit are retained at zero magnetic field in contrast to other spin qubit implementations, the quantum device architecture used in this work is compatible for realizing a high fidelity transmon–spin-qubit and spin-qubit–spin-qubit interface in a future quantum processor.

## Methods

**RX qubit tunnel coupling configurations.** Throughout this work, we use the four RX qubit tunnel coupling configurations listed in Table 1.

The measurements to extract the tunnel couplings are explained in Supplementary Note 1. Different configurations were necessary for two reasons. To realize the virtual interaction scheme in Fig. 3c for different RX qubit working points while keeping the same transmon flux, the tunnel couplings had to be adjusted. The current in the transmon flux line was kept below a level at which an increase in the refrigerator temperature was observed. This ensured that the device operation took place at the lowest accessible measurement temperature. Second, when readjusting the RX qubit after a random charge rearrangement occurred in the host material, which was observed on the time scale of days, identical tunnel coupling configurations could not be achieved.

**Details of sample and measurement scheme.** In Fig. 4a we show a false-colored optical micrograph of the part of the sample that was illustrated schematically in Fig. 1a. The microwave read-out scheme is also indicated. The sample is measured in a dilution refrigerator with a base temperature of 10 mK. The four quantum systems are highlighted in different colors. A magnified image of the transmon is shown in Fig. 4b. One side of the SQUID loop is grounded, the other is connected to a big shunt capacitor (highlighted in green). We control the transmon transition frequency with a current $I$ though an inductively coupled flux line. The transmon is capacitively coupled to one end of a $\lambda/2$ 50 $\Omega$ (read-out) resonator, which is shown

to the full extent in Fig. 4c. It is capacitively coupled to a transmission line that is used for resonator read-out.

## Data availability

The data of this study are available from the corresponding author on reasonable request.

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

| Table 1 Tunnel couplings | | |
|---|---|---|
| RX qubit config. | $t_l/h$ (GHz) | $t_r/h$ (GHz) |
| 1 | 9.91 | 8.26 |
| 2 | 9.22 | 8.73 |
| 3 | 8.52 | 8.18 |
| 4 | 8.80 | 8.77 |
| RX qubit tunnel coupling configurations used in this work | | |

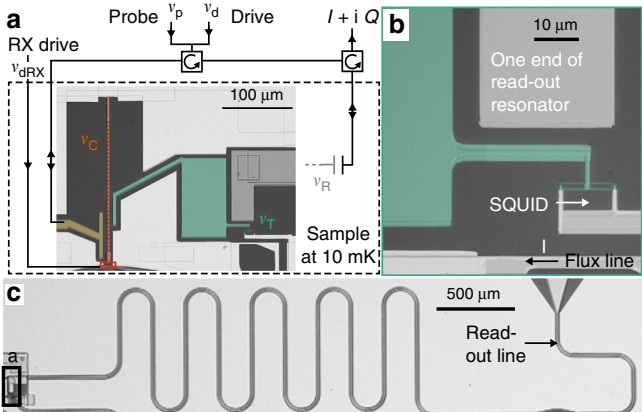

**Fig. 4** Sample details. False-colored optical micrographs. **a** Relevant part of the sample (dashed region) false colored as follows: SQUID array (coupling) resonator in orange with corresponding drive and probe port in yellow, transmon in green, one end of the 50 $\Omega$ (read-out) resonator in gray, GaAs in black and grounded Al in white. The position of the TQD (see Fig. 1c) is outlined with a red rectangle. The microwave lines that route probe and drive tones to the sample are shown. The frequencies $\nu_i$ are labeled as in Fig. 1a. **b** Magnified optical image of the transmon region in **a**. The positions of transmon SQUID and flux line with current $I$ are marked. **c** Optical micrograph of the 50 $\Omega$ resonator and its microwave read-out line. The location of the region shown in **a** is outlined with a black rectangle

30. Koch, J. et al. Charge-insensitive qubit design derived from the Cooper pair box. *Phys. Rev. A* **76**, 042319 (2007).
31. Houck, A. A. et al. Controlling the spontaneous emission of a superconducting transmon qubit. *Phys. Rev. Lett.* **101**, 080502 (2008).
32. Frey, T. et al. Dipole coupling of a double quantum dot to a microwave resonator. *Phys. Rev. Lett.* **108**, 046807 (2012).
33. Schuster, D. I. et al. ac Stark shift and dephasing of a superconducting qubit strongly coupled to a cavity field. *Phys. Rev. Lett.* **94**, 123602 (2005).
34. Schreier, J. A. et al. Suppressing charge noise decoherence in superconducting charge qubits. *Phys. Rev. B* **77**, 180502 (2008).
35. Russ, M. & Burkard, G. Three-electron spin qubits. *J. Phys. Condens. Matter* **29**, 393001 (2017).
36. Tokura, Y., vanderWiel, W. G., Obata, T. & Tarucha, S. Coherent single electron spin control in a slanting zeeman field. *Phys. Rev. Lett.* **96**, 047202 (2006).
37. Pioro-Ladrière, M. et al. Electrically driven single-electron spin resonance in a slanting Zeeman field. *Nat. Phys.* **4**, 776–779 (2008).
38. Golovach, V. N., Borhani, M. & Loss, D. Electric-dipole-induced spin resonance in quantum dots. *Phys. Rev. B* **74**, 165319 (2006).
39. Nowack, K. C., Koppens, F. H. L., Nazarov, Y. V. & Vandersypen, L. M. K. Coherent control of a single electron spin with electric fields. *Science* **318**, 1430–1433 (2007).
40. Cottet, A. & Kontos, T. Spin quantum bit with ferromagnetic contacts for circuit QED. *Phys. Rev. Lett.* **105**, 160502 (2010).
41. Viennot, J. J., Dartiailh, M. C., Cottet, A. & Kontos, T. Coherent coupling of a single spin to microwave cavity photons. *Science* **349**, 408–411 (2015).
42. Gambetta, J. et al. Qubit-photon interactions in a cavity: measurement-induced dephasing and number splitting. *Phys. Rev. A* **74**, 042318 (2006).
43. Sete, E. A., Gambetta, J. M. & Korotkov, A. N. Purcell effect with microwave drive: suppression of qubit relaxation rate. *Phys. Rev. B* **89**, 104516 (2014).
44. Malinowski, F. K. et al. Symmetric operation of the resonant exchange qubit. *Phys. Rev. B* **96**, 045443 (2017).
45. Johnson, A. C. et al. Triplet-singlet spin relaxation via nuclei in a double quantum dot. *Nature* **435**, 925–928 (2005).
46. Koppens, F. H. L. et al. Control and detection of singlet-triplet mixing in a random nuclear field. *Science* **309**, 1346–1350 (2005).
47. Koppens, F. H. L. et al. Driven coherent oscillations of a single electron spin in a quantum dot. *Nature* **442**, 766–771 (2006).
48. Trauzettel, B., Bulaev, D. V., Loss, D. & Burkard, G. Spin qubits in graphene quantum dots. *Nat. Phys.* **3**, 192–196 (2007).
49. Eich, M. et al. Spin and valley states in gate-defined bilayer graphene quantum dots. *Phys. Rev. X* **8**, 031023 (2018).
50. Zwanenburg, F. A. et al. Silicon quantum electronics. *Rev. Mod. Phys.* **85**, 961–1019 (2013).
51. Eng, K. et al. Isotopically enhanced triple-quantum-dot qubit. *Sci. Adv.* **1**, e1500214 (2015).
52. Andrews, R. W. et al. Quantifying error and leakage in an encoded Si/SiGe triple-dot qubit. *Nat. Nano* **14**, 747–750 (2019).

## Acknowledgements

We acknowledge discussions with Guido Burkard, Michele Collodo, Christian Kraglund Andersen and Maximilian Russ. We also thank David van Woerkom for his contribution to the sample fabrication and for input to the paper. This work was supported in part by the Swiss National Science Foundation through the National Center of Competence in Research (NCCR) Quantum Science and Technology and the project Elements for Quantum Information Processing with Semiconductor/Superconductor Hybrids (EQUIPS). S.N.C. and M.F. acknowledge support of the Vannevar Bush Faculty Fellowship program sponsored by the Basic Research Office of the Assistant Secretary of Defense for Research and Engineering and the funding by the Office of Naval Research through Grant No. N00014-15-1-0029. M.F. and J.C.A.U. acknowledge the support by ARO (W911NF-17-1-0274). The views and conclusions contained in this work are those of the authors and should not be interpreted as necessarily representing the official policies or endorsements, either expressed or implied, of the Army Research Office (ARO) or the U.S. Government. The U.S. Government is authorized to reproduce and distribute reprints for Governmental purposes, notwithstanding any copyright notation thereon.

## Author contributions

A.J.L., J.V.K. and P.S. fabricated the sample. A.J.L. and J.V.K. performed the measurements with input from B.K. A.J.L. analyzed the data. A.J.L., C.M. and J.C.A.U. wrote the paper with input from all authors. C.R. grew the heterostructure under the supervision of W.W. C.M. developed the circuit QED theory. J.C.A.U. derived the hyperfine noise model under the supervision of S.N.C. and M.F. A.W., T.I. and K.E. supervised the experiment.

## Competing interests

The authors declare no competing interests.
