## [Peer Review File · Nature Communications]

Reviewers' comments:

Reviewer #1 (Remarks to the Author):

The manuscript by A. J. Landig et.al. presents an interesting demonstration experiment regarding the coherent long-distance coupling between a spin qubit and a superconducting qubit. The authors fabricated a chip with both a triple-quantum-dot spin qubit and a transmon strongly coupled to a tunable resonator. With this hybrid quantum circuit, they realized the indirect interaction between the spin qubit and the transmon qubit via a resonator.

The strong coupling regime is not realized yet, mainly due to the hyperfine interaction in the material, according to the authors' statement. This is not as good as some of the other hybrid systems, but the experiment was well performed and the manuscript is well organized and also contains important results. Therefore, I think it is of the novelty and interest for the NC readership.

Below I list some comments for the authors to consider:

1. In the abstract, the authors state that 'We spectroscopically observe coherent interaction between the resonant exchange qubit and a transmon qubit in both resonant and dispersive regimes', but as we can see in fig.2(d), the interaction in the resonant regime is actually observed between the hybrid state of the transmon-resonator system and the RX qubit, or the resonant interaction between the resonator and the qubits, not really the direct interaction between the qubits. In this sense, it is another type of indirect interaction, as compared with the indirect interaction between the two qubits via the virtual resonator photons. Thus, an appropriate statement is needed for this in the abstract and the main text, so as to avoid confusion.
2. The decoherence of the quantum-dot system used in the present manuscript is limited by the hyperfine interaction in the GaAs host material. What is the main advantage of using the present quantum-dot system, instead of using the hyperfine free materials? Are there any technical handicaps when using the latter? Some brief discussions are useful to readers.
3. As I know, different cases of the interactions between a spin qubit and a superconducting qubit were discussed in Rev. Mod. Phys. 85, 623 (2013). It could be properly cited because the two types of indirect interactions via a resonator, as studied in the present manuscript, were also discussed there.

Reviewer #2 (Remarks to the Author):

This paper demonstrates via spectroscopy the coupling of a transmon qubit to a RX GaAs triple quantum dot via a high-impedance tunable microwave resonator. They independently demonstrate individual resonant coupling of the two qubits to the resonator as a function of qubit frequency and then couple the two qubits to one another, when either resonant with the cavity or off-resonance (with "virtual" photons). They also demonstrate that tuning the charge character of the RX qubit (via the parameter Δ) affects its coherence (being a charge qubit = worse), and show that there is an interplay between calculated g-factor and Δ . Many of the data are accompanied by master equation simulations that have been parameterized by other measurements, showing good agreement.

This paper is a direct follow-on to two previous papers from ETH (both with a very similar author lists). The first is available in a pre-print at arXiv:1806.10039. There, they couple a double-dot charge qubit to a transmon using a nearly identical apparatus. This work actually had better coupling rates (66 vs. 52 MHz) and dephasing rates (3 MHz vs. at best, 6.5 MHz), but is a charge

qubit and so naturally benefits from a greater dipole moment and insensitivity to nuclear noise. This paper also demonstrated a time-resolved coherent oscillation of an excitation between the DQD and transmon. The second paper, Nature 560, 179 (2018), demonstrated strong coupling of a RX qubit to a high impedance (non-tunable) microwave resonator. This result showed worse coupling (23 MHz) and dephasing (10 MHz) than the result under review, but carefully measured the microwave coupling strength as a function of Δ using the AC stark shift.

The claims of the paper under consideration here are:

- * Coherent coupling between a spin qubit and a transmon
 - * That this coupling is "long distance"
 - * That this is important because they're coupling a transmon to the spin without a magnetic field
- Let's evaluate the claims one-by-one.

First, they claim the coupling between the spin qubit and the transmon is coherent. Though I suppose there isn't direct experimental evidence (just agreement with theory), I believe that the triple quantum dot operated as an RX qubit can (at least in some regimes) be considered a spin qubit. I also believe the superconducting qubit is a transmon. The spectroscopy shows the two systems tuning independently and as predicted, and so the avoided crossing between the two qubits is convincing evidence that the interaction is a coherent one. The claims are also strongly supported by the theoretical simulations (which themselves are thoroughly documented in the supplement).

Though this (relatively modest) claim of coherence is satisfactorily supported by the evidence, I would like to register some level of disappointment. There are no time-resolved measurements of either the coherence of the transmon or RX qubits or of interactions of the two together. In particular, I'm interested in whether this transmon is substantially degraded by the quantum dot fabrication process or materials. They explain in their Nature paper that the 2DEG is etched away prior to fabricating the superconducting components. Is this effective at returning the coherence to its expected level? Perhaps the answer to this question is obscured by the fact that the transmon is Purcell-limited near the cavity and limited by charge dispersion away from it, but if that's the case, it could be mentioned. (Or perhaps GaAs is a bad host material, but using Si would be expected to improve things substantially.) The context of this paper is that it could be useful for spin qubits to couple over long distances or for a transmon to have a spin memory. If the act of adding that spin irrevocably destroys the properties of the transmon, it undermines the whole purpose.

Additionally, the lack of time-resolved data of the interaction between the RX and transmon makes me concerned. Is there some reason this experiment couldn't be conducted? It was done quite straightforwardly in the previous paper; it only involves fluxing the superconducting qubit non-adiabatically, a technique that has been used extensively by the superconducting community for a decade. Did the fridge not afford fast flux control? Does pulsing the flux line unacceptably warm the quantum dot? (Etc?)

Perhaps related to that question, the authors say in the Methods that "the transmon flux line current was kept low enough to exclude the potential heating of the sample" and that "when readjusting the RX qubit after an electrostatic jump, identical tunnel coupling configurations could not be achieved." The way this paragraph is written is very hard for me to follow. What electrostatic jump? All of the data in this paper are CW; there is no fast electrostatic jumping. Does moving the qubit(s) slowly cause hysteresis, perhaps by locking in vortices in the superconductor? Does the heating associated with the transmon flux prevent fast-flux pulses from being used? If so, why were they permissible in the previous paper? Answering all these questions in particular is not important; rather, this paragraph should be clarified and expanded. I simply do not understand the point it is trying to make and to what extent, if at all, these issues affected this experiment and potential future experiments.

Nevertheless, I'm fine with the claim that there is a coherent coupling between a spin qubit and a transmon.

Second, the authors claim this coupling is "long distance." They themselves define a long distance

coupling as one which "couples both qubit systems controllably over distances exceeding the physical size of the qubit [...] by several orders of magnitude" (note the typo "ordes" in the submitted text). They later say that the "transmon and the RX qubit are capacitively coupled to the same end of a SQUID array resonator" which places the RX qubit "at a distance of a few hundred micrometers from the transmon". Considering that the transmon itself is several hundred microns in dimension, calling this coupling "long distance" is extremely dubious. Perhaps if *both* qubits both had dimensions of "hundreds of nanometers" (as the authors hopefully suggest), this coupling could be considered long distance. But since one of the qubits is essentially the same dimension as the distance between the two, I don't buy it.

Of course, there are other (very similar) geometries that would very likely support this claim, perhaps by placing the two qubits on opposite ends of a $\lambda/2$ resonator (e.g. as shown in the related Nature paper). It also may be true that the direct capacitive coupling of the transmon to the TQD is totally negligible. But that's not what they do here, and I do not believe this claim is supported by the paper (even though it easily could've been, with a design change). In reality, the resonator in this experiment is just playing the role of an impedance transformer between the low-impedance transmon and the weak dipole moment of the RX qubit. The distributed nature of the resonator is totally unimportant.

I should mention, however, that I think the claim of "long distance" is relatively unimportant in evaluating the merit of this paper, for the reasons stated above.

Third, the authors claim that the point of this work is that it demonstrates coupling of a spin qubit to a transmon, which is hard because coupling a spin to a microwave normally requires a magnetic field. It is true that the RX qubit is interesting because it can produce a direct electric dipole to the spin degree of freedom of trapped electrons. The hope is that you could pulse Δ to turn on a transient dipole moment (by being more charge-like) and do your microwave interactions, then pulse it back to a large negative value (hence making the qubit spin-like, but with a very weak dipole moment) where coherence should be far better to do single-qubit gate operations or just to use as a memory. Unfortunately, this benefit is not so obvious in GaAs due to the noisy nuclear spin bath, but there's no reason this system can't be a test bench for other quieter materials (as the authors mention in their conclusion). So yes: RX is an excellent encoding for coupling the spin of a quantum dot array to a single microwave photon, be it an excitation of a microwave resonator or transmon junction.

However, in the context of the related Nature paper, I have serious doubts about whether this claim is novel. The same authors just demonstrated strong coupling of an RX qubit to a high-impedance microwave cavity. The fact that a superconducting qubit coupled to the same cavity behaves exactly as the Jaynes-Cummings hamiltonian would predict it would is not in any way surprising. (Coupling a superconducting qubit to a microwave cavity has been a standard technique for fifteen years! And a transmon specifically for twelve.) The scientific questions you'd be concerned about a priori -- for example, whether the coherence of the transmon is disrupted by the semiconductor qubit host materials -- are (as far as I am aware) not touched on by this or the related arXiv paper.

I am therefore inclined to view the fact that a transmon is present in this device as nearly scientifically irrelevant. There simply is no reason to expect, given the two related papers, that this coupling wouldn't work exactly the way that it does here. If the authors perhaps demonstrated a two-qubit gate between the qubits (a goal no doubt they are working toward) or showed any sort of non-trivial interaction between the qubits, my disposition would of course be different. But the difficult job of strong coupling of an RX qubit to a microwave was already accomplished (and richly rewarded); the transmon works just as expected *and* previously demonstrated. By the same group.

There are, however, some noteworthy innovations in this demonstration compared to the related Nature paper. First, they have incorporated a tunable microwave resonator. They are able to use this to more thoroughly characterize the system, though they briefly mention that it adds extra charge noise. It'd be nice to hear more about what's going on there, but perhaps that will require future work (if not, please say so). Second, and far more importantly, they have substantially

improved the quality of the RX qubit. The g factor is much higher than the related Nature result (52 MHz vs. 23 MHz) and the dephasing rate much lower (6.5 MHz vs. 10 MHz). This improvement is substantial and will be key to demonstrating a two-qubit gate. Unfortunately, as far as I am aware, the authors make no note of this fact nor offer any reasons for its occurrence. The resonator impedance has not increased since the Nature result (perhaps now somewhat smaller). Is the device tune-up or operation superior? The geometry improved? Materials or interfaces cleaner? (Etc.?) Since I view this as the primary scientific achievement of this paper, I would strongly urge the authors to expand on the reasons for this improvement to the extent that they are aware (or can speculate).

If I were revising this paper in light of reviewer criticism such as written above, I would consider advertising the fact that the coupling strength and dephasing improvements enable higher-quality coupling to the transmon. Something like "this second-order coupling (e.g. RX to tmon) places greater demands on the RX qubit, etc. etc." While this isn't quite as flashy, it has the virtue of actually being true, since the goal is to eventually entangle the two qubits. It's also obviously the kind of information that would be of interest to others in the community.

On balance, I feel that there is important work present in this paper and I lean toward recommending publication after substantial revision. Coupling semiconductor spin qubits to microwave resonators is a very active current research direction and these authors are among the world leaders. However, not every result should be advertised as a breathtaking breakthrough, and I think the authors have missed the mark on selling this one. They should dial back on the grandiosity of their claims and focus more on the technical improvements demonstrated here, which are much more meaningful.

-Dr. Matthew Reed
HRL Labs

Response to referee comments on manuscript “Coherent long-distance spin-qubit-transmon coupling”

Reviewer 1

Comment 1:

The manuscript by A. J. Landig et.al. presents an interesting demonstration experiment regarding the coherent long-distance coupling between a spin qubit and a superconducting qubit. The authors fabricated a chip with both a triple-quantum-dot spin qubit and a transmon strongly coupled to a tunable resonator. With this hybrid quantum circuit, they realized the indirect interaction between the spin qubit and the transmon qubit via a resonator. The strong coupling regime is not realized yet, mainly due to the hyperfine interaction in the material, according to the authors' statement. This is not as good as some of the other hybrid systems, but the experiment was well performed and the manuscript is well organized and also contains important results. Therefore, I think it is of the novelty and interest for the NC readership.

Response 1:

We thank the referee for the positive assessment of our work. As pointed out by them, hyperfine interaction limits the RX qubit performance. However despite the hyperfine interaction, the strong coupling limit between the RX qubit and the resonator photons is in fact realized as indicated by the vacuum Rabi mode splitting in Fig. 2c. Furthermore, the extracted qubit-photon coupling strength $g_{RX}/2\pi \approx 52$ MHz clearly exceeds both the qubit decoherence rate $\gamma_{2,RX} \approx 11$ MHz and the cavity photon decay rate $\kappa_C/2\pi \approx 4.6$ MHz. In addition, the maximum ratio of coupling strength to decoherence rate is comparable to state-of-the-art values reached with Si spin qubits in circuit QED. We kindly ask the referee to point our attention to misleading statements in our manuscript that led to their conclusion “the strong coupling regime is not realized yet”.

Comment 2:

In the abstract, the authors state that ‘We spectroscopically observe coherent interaction between the resonant exchange qubit and a transmon qubit in both resonant and dispersive regimes’, but as we can see in fig.2(d), the interaction in the resonant regime is actually observed between the hybrid state of the transmon-resonator system and the RX qubit, or the resonant interaction between the resonator and the qubits, not really the direct interaction between the qubits. In this sense, it is another type of indirect interaction, as compared with the indirect interaction between the two qubits via the virtual resonator photons. Thus, an appropriate statement is needed for this in the abstract and the main text, so as to avoid confusion.

Response 2:

As pointed out by the referee, the RX qubit interacts with the hybridized transmon-resonator states $|\pm\rangle$ at the well pronounced avoided crossings in Fig. 2d at $\Delta/h \approx -9.8$ GHz and $\Delta/h \approx -5.6$ GHz. In between these avoided crossings at $\Delta/h \approx -7.8$ GHz, the RX qubit, the transmon and the resonator are on resonance ($\nu_{RX} = \nu_T = \nu_C$). There, the splitting of the dips in the reflection spectrum is enhanced by ≈ 16 MHz compared to the bare transmon-resonator splitting between $|\pm\rangle$ (see $\Delta/h \approx -11.4$ GHz in Fig. 2d). We calculate the theoretical prediction $(2g - 2g_T)/2\pi \approx 15$ MHz with the collective coupling

strength $g/2\pi = \sqrt{(g_T^2 + g_{RX}^2)}$ and $g_T/2\pi = 180$ MHz as well as $g_{RX}/2\pi = 52$ MHz from Figs. 2a-b. Theoretical and experimental enhancement are in good agreement and we conclude, that we observe an experimental signature of the coherent resonant interaction of all three quantum systems. We note that this enhancement is small since $g_{RX} \ll g_T$ and therefore barely visible in Fig. 2d (see separation of black and red dashed line at $\Delta/h \approx -7.8$ GHz). We therefore show the experimental cuts extracted from Fig. 2d at $\Delta/h \approx -7.8$ GHz and $\Delta/h \approx -11.4$ GHz (bare transmon-resonator splitting) in the new Supplementary Fig. 4.

We have modified the manuscript as follows: We have added the paragraph “The RX qubit, transmon and [...] parameters.” on page 4. We have also marked $\Delta/h \approx -7.8$ GHz in Fig. 2d and modified the figure caption correspondingly. In addition, we added the subsection “Resonant RX qubit-transmon-resonator” interaction to the Supplementary Information.

Comment 3:

The decoherence of the quantum-dot system used in the present manuscript is limited by the hyperfine interaction in the GaAs host material. What is the main advantage of using the present quantum-dot system, instead of using the hyperfine free materials? Are there any technical handicaps when using the latter? Some brief discussions are useful to readers.

Response 3:

As pointed out by the referee, the hyperfine interaction in GaAs imposes a limit on the qubit coherence. Alternative material systems with minimal hyperfine interaction are graphene and silicon. GaAs was still the material of choice for our proof-of-principle experiments for several reasons. First, GaAs allows a high level of control over the electronic states in the quantum dots. This is necessary in order to realize resonant exchange qubit energies of the order of 4GHz, which require high tunnel couplings simultaneously between the left and the middle and the middle and the right quantum dot of about 8GHz. Achieving such a tunnel coupling configuration is potentially more challenging in silicon based quantum dots, which are roughly a factor of two smaller due to the larger effective mass compared to GaAs and require multiple gate layers for electron accumulation and depletion. To the best of our knowledge, a resonant exchange qubit with a sufficiently high qubit energy for circuit QED has not yet been implemented in Si. Third, in contrast to GaAs, the valley degree of freedom can become relevant for graphene or silicon and can, for instance, introduce additional leakage channels for the qubit.

Following the suggestion of the referee, we have added the following paragraph to the conclusion: “While three-electron [...] degree of freedom [Russ et al. 2017].” This paragraph includes the new references Eng et al. 2015 and Andrews et al. 2018 (preprint).

Comment 4:

As I know, different cases of the interactions between a spin qubit and a superconducting qubit were discussed in Rev. Mod. Phys. 85, 623 (2013). It could be properly cited because the two types of indirect interactions via a resonator, as studied in the present manuscript, were also discussed there.

Response 4:

We thank the referee for pointing our attention to this paper.

We have added a reference to the introduction: “[...] interface between spin and superconducting qubits [Xiang et al. 2013] [...]”. We also added two additional references to related theoretical studies to the introduction: “[...] spin qubit to another distant qubit [Srinivasa et al. 2016, Benito et al. 2019] [...]”.

Reviewer 2

Comment 1:

First, they claim the coupling between the spin qubit and the transmon is coherent. Though I suppose there isn't direct experimental evidence (just agreement with theory), I believe that the triple quantum dot operated as an RX qubit can (at least in some regimes) be considered a spin qubit. I also believe the superconducting qubit is a transmon. The spectroscopy shows the two systems tuning independently and as predicted, and so the avoided crossing between the two qubits is convincing evidence that the interaction is a coherent one. The claims are also strongly supported by the theoretical simulations (which themselves are thoroughly documented in the supplement).

Though this (relatively modest) claim of coherence is satisfactorily supported by the evidence, I would like to register some level of disappointment. There are no time-resolved measurements of either the coherence of the transmon or RX qubits or of interactions of the two together. In particular, I'm interested in whether this transmon is substantially degraded by the quantum dot fabrication process or materials. They explain in their Nature paper that the 2DEG is etched away prior to fabricating the superconducting components. Is this effective at returning the coherence to its expected level? Perhaps the answer to this question is obscured by the fact that the transmon is Purcell-limited near the cavity and limited by charge dispersion away from it, but if that's the case, it could be mentioned. (Or perhaps GaAs is a bad host material, but using Si would be expected to improve things substantially.) The context of this paper is that it could be useful for spin qubits to couple over long distances or for a transmon to have a spin memory. If the act of adding that spin irrevocably destroys the properties of the transmon, it undermines the whole purpose.

Response 1:

As pointed out by the referee, we find the transmon in our experiment to be Purcell limited and therefore cannot make a statement about the influence of the transmon substrate material on its coherence properties. Our study puts the focus on the coherence of the RX qubit, which is the bottleneck in terms of coherence.

Comment 2:

Additionally, the lack of time-resolved data of the interaction between the RX and transmon makes me concerned. Is there some reason this experiment couldn't be conducted? It was done quite straightforwardly in the previous paper; it only involves fluxing the superconducting qubit non-adiabatically, a technique that has been used extensively by the superconducting community for a decade. Did the fridge not afford fast flux control? Does pulsing the flux line unacceptably warm the quantum dot? (Etc?)

Response 2:

In the experiment by Scarlino et al., mentioned by the referee, the ratio of the exchange splitting $2J$ over the charge qubit decoherence rate was about 7.8. In our experiment, we obtain 3.3 for this ratio at the optimal RX qubit working point (see green line in Fig. 3h and e) in Supplementary Table 9), which would wash out the coherent oscillations (about one oscillation would be visible) even more compared to the data in Scarlino et al. Hence there was no strong incentive to prepare our measurement setup at hand, which allows for spectroscopy measurements, for time-resolved experiments.

Comment 3:

Perhaps related to that question, the authors say in the Methods that “the transmon flux line current was kept low enough to exclude the potential heating of the sample” and that “when readjusting the RX qubit after an electrostatic jump, identical tunnel coupling configurations could not be achieved.” The way this paragraph is written is very hard for me to follow. What electrostatic jump? All of the data in this paper are CW; there is no fast electrostatic jumping. Does moving the qubit(s) slowly cause hysteresis, perhaps by locking in vortices in the superconductor? Does the heating associated with the transmon flux prevent fast-flux pulses from being used? If so, why were they permissible in the previous paper? Answering all these questions in particular is not important; rather, this paragraph should be clarified and expanded. I simply do not understand the point it is trying to make and to what extent, if at all, these issues affected this experiment and potential future experiments.

Response 3:

It turned out in our experiment, that the current necessary to tune the transmon by one flux quantum caused heating of the refrigerator base plate by a few mK. To ensure the lowest accessible experimental temperature, the transmon flux was therefore kept below values where such a temperature increase could be observed. This issue can be avoided in future experiments by lowering the required current with a design change of the on-chip flux line. Note, that this issue was not the reason for omitting time-resolved measurements as pointed out above.

The term “electrostatic jump” refers to a random charge rearrangement in the heterostructure (RX qubit) that occurs on the time scale of days and necessitates to retune the qubit electrostatically to the desired working point. After retuning the qubit, the tunnel coupling configurations were not identical compared to before the charge rearrangement. This is a standard issue for most semiconductor-based quantum dots.

Following the suggestion of the referee, we have modified the methods section “RX tunnel coupling configurations”: “The current in the [...] be achieved”.

Comment 4:

*Second, the authors claim this coupling is “long distance.” They themselves define a long distance coupling as one which “couples both qubit systems controllably over distances exceeding the physical size of the qubit [...] by several orders of magnitude” (note the typo “ordes” in the submitted text). They later say that the “transmon and the RX qubit are capacitively coupled to the same end of a SQUID array resonator” which places the RX qubit “at a distance of a few hundred micrometers from the transmon”. Considering that the transmon itself is several hundred microns in dimension, calling this coupling “long distance” is extremely dubious. Perhaps if *both* qubits both had dimensions of “hundreds of nanometers” (as the authors hopefully suggest), this coupling could be considered long distance. But since one of the qubits is essentially the same dimension as the distance between the two, I don’t buy it. Of course, there are other (very similar) geometries that would very likely support this claim, perhaps by placing the two qubits on opposite ends of a $\lambda/2$ resonator (e.g. as shown in the related Nature paper). It also may be true that the direct capacitive coupling of the transmon to the TQD is totally negligible. But that’s not what they do here, and I do not believe this claim is supported by the paper (even though it easily could’ve been, with a design change). In reality, the resonator in this experiment is just playing the role of an impedance transformer between the low-impedance transmon and the weak dipole moment of the RX qubit. The distributed nature of the resonator is totally unimportant. I should mention, however, that I think the claim of “long distance” is relatively unimportant in*

evaluating the merit of this paper, for the reasons stated above.

Response 4:

We agree with the referee that the term “long-distance” can be misinterpreted due to the significantly larger size of the transmon compared to the RX qubit.

Following the suggestion of the referee, we decided to change the title to “Virtual-photon-mediated spin-qubit-transmon coupling” and removed “long-distance” in the abstract, the introduction as well as in the conclusion. In addition, we modified the following sentence at the bottom of page 2: “At a distance of a few hundred micrometers from the transmon SQUID, we form [...]” We also corrected the typo in “orders”.

Comment 5:

Third, the authors claim that the point of this work is that it demonstrates coupling of a spin qubit to a transmon, which is hard because coupling a spin to a microwave normally requires a magnetic field. It is true that the RX qubit is interesting because it can produce a direct electric dipole to the spin degree of freedom of trapped electrons. The hope is that you could pulse Δ to turn on a transient dipole moment (by being more charge-like) and do your microwave interactions, then pulse it back to a large negative value (hence making the qubit spin-like, but with a very weak dipole moment) where coherence should be far better to do single-qubit gate operations or just to use as a memory. Unfortunately, this benefit is not so obvious in GaAs due to the noisy nuclear spin bath, but there’s no reason this system can’t be a test bench for other quieter materials (as the authors mention in their conclusion).

So yes: RX is an excellent encoding for coupling the spin of a quantum dot array to a single microwave photon, be it an excitation of a microwave resonator or transmon junction.

However, in the context of the related Nature paper, I have serious doubts about whether this claim is novel. The same authors just demonstrated strong coupling of an RX qubit to a high-impedance microwave cavity. The fact that a superconducting qubit coupled to the same cavity behaves exactly as the Jaynes-Cummings hamiltonian would predict it would is not in any way surprising. (Coupling a superconducting qubit to a microwave cavity has been a standard technique for fifteen years! And a transmon specifically for twelve.) The scientific questions you’d be concerned about a priori -- for example, whether the coherence of the transmon is disrupted by the semiconductor qubit host materials -- are (as far as I am aware) not touched on by this or the related arXiv paper.

*I am therefore inclined to view the fact that a transmon is present in this device as nearly scientifically irrelevant. There simply is no reason to expect, given the two related papers, that this coupling wouldn’t work exactly the way that it does here. If the authors perhaps demonstrated a two-qubit gate between the qubits (a goal no doubt they are working toward) or showed any sort of non-trivial interaction between the qubits, my disposition would of course be different. But the difficult job of strong coupling of an RX qubit to a microwave was already accomplished (and richly rewarded); the transmon works just as expected **and** previously demonstrated. By the same group.*

Response 5:

The referee is correct in pointing out the relation of our work to many other works before. We obviously agree that coupling a transmon to a resonator has been done before. In general, the coupling of a transmon to a spin qubit is not a straight forward task at all in spite of the fact that all experimental results came out as expected. As pointed out by the referee, the coupling of a spin qubit to a transmon is challenging due to the magnetic field sensitivity of the latter qubit. Other spin qubit systems that have been implemented on the circuit QED platform (Mi et al. 2018, Samkharadze et al. 2018) require external magnetic fields of a few hundred millitesla as well as magnetic materials in the vicinity of the

spin qubit. In our previous work, mentioned by the referee, the RX qubit was implemented in a magnetic field of 200 mT. In the present work, we demonstrate experimentally and confirm theoretically, that we obtain the same qubit coherence in the absence of a magnetic field (see response 7), which is essential to interface the RX qubit with a transmon qubit. Consequently, we consider the presence of the transmon on the same chip as the spin qubit to be an important message of this paper. Thanks to the referee, we became aware of two pioneering studies of an exchange based three-electron qubit in Si (Andrews et al. 2018 (preprint), Eng et al. 2015) at zero magnetic field with charge sensor based read-out, which however do not compare the coherence performance with operation in a finite magnetic field.

We added a reference to both studies to the conclusion (see response 3 to reviewer 1).

Comment 6:

There are, however, some noteworthy innovations in this demonstration compared to the related Nature paper. First, they have incorporated a tunable microwave resonator. They are able to use this to more thoroughly characterize the system, though they briefly mention that it adds extra charge noise. It'd be nice to hear more about what's going on there, but perhaps that will require future work (if not, please say so).

Response 6:

We thank the referee for pointing out this issue. We speculate that TLSs in the resonator junctions introduce charge noise. To make a reliable claim, further investigation is required in a dedicated experimental setup.

Comment 7:

Second, and far more importantly, they have substantially improved the quality of the RX qubit. The g factor is much higher than the related Nature result (52 MHz vs. 23 MHz) and the dephasing rate much lower (6.5 MHz vs. 10 MHz). This improvement is substantial and will be key to demonstrating a two-qubit gate. Unfortunately, as far as I am aware, the authors make no note of this fact nor offer any reasons for its occurrence. The resonator impedance has not increased since the Nature result (perhaps now somewhat smaller). Is the device tune-up or operation superior? The geometry improved? Materials or interfaces cleaner? (Etc.?) Since I view this as the primary scientific achievement of this paper, I would strongly urge the authors to expand on the reasons for this improvement to the extent that they are aware (or can speculate). If I were revising this paper in light of reviewer criticism such as written above, I would consider advertising the fact that the coupling strength and dephasing improvements enable higher-quality coupling to the transmon. Something like "this second-order coupling (e.g. RX to tmon) places greater demands on the RX qubit, etc. etc." While this isn't quite as flashy, it has the virtue of actually being true, since the goal is to eventually entangle the two qubits. It's also obviously the kind of information that would be of interest to others in the community.

Response 7:

We thank the referee for pointing out these achievements in our manuscript. Indeed, the qubit-photon coupling strength in this work has roughly doubled compared to our previous work possibly for the following reasons. First, the characteristic impedance of the resonator in the present work is at the qubit energy ($Z_C \approx 1.8 \text{ k}\Omega$ at $\nu_C = 4.2 \text{ GHz}$ since $Z_C \propto 1/\nu_C$) enhanced by a factor of ≈ 1.4 compared to the resonator in our earlier work. Second, in this experiment we position the TQD directly at the resonator end and therefore at the maximum of the voltage fluctuations, while there was a spatial offset from this maximum in our previous study. Third, the design of the TQD fine gates was modified in the present

work in order to enlarge the overlap of the resonator gate with the underlying quantum dot. The increased capacitance between the gate and the dot enhances the resonator lever arm.

We agree with the referee, that while in our previous work an averaged minimum qubit decoherence rate of about 10MHz was measured, we now obtain about 6.5MHz. This is due the significantly larger measurement range in Δ in our present work, which is facilitated by the enhanced qubit-photon coupling strength. Consequently, while there was still a remaining contribution due to charge noise in our previous study (at $\Delta/h \approx -10\text{GHz}$), hyperfine-induced dephasing fully dominates the decoherence rate for the data in Fig. 3a at $\Delta/h \ll -10\text{GHz}$ and $\gamma_{2,\text{RX}}(\Delta)$ becomes flat. If we use the noise model of our current study to fit to the data of our previous work, we obtain within the error range the same width σ_B of the Overhauser field fluctuations.

Following the suggestion by the referee, we have modified the first paragraph in the subsection “Resonant interaction” on page 3 as follows: “This coupling strength [...] underlying quantum dot.”

Comment 8:

On balance, I feel that there is important work present in this paper and I lean toward recommending publication after substantial revision. Coupling semiconductor spin qubits to microwave resonators is a very active current research direction and these authors are among the world leaders. However, not every result should be advertised as a breathtaking breakthrough, and I think the authors have missed the mark on selling this one. They should dial back on the grandiosity of their claims and focus more on the technical improvements demonstrated here, which are much more meaningful.

Response 8:

We thank the referee for his positive assessment of our experiment, and hope that the revised manuscript more properly addresses the practical achievements without unwarranted advertisement.

Comments for the editor

- The labels of the color scale in Supplementary Figure 2 were corrected.
- Ref. 29 was updated.
- A funding source was added.
- The data availability statement was added.
- The definition of $|0\rangle$ and $|1\rangle$ in Eq. (2) in Supplementary Information was corrected. Consequently, Eqns. (3) and (20) were adjusted.
- A missing subscript in Eq. (14) in Supplementary Information was added.

REVIEWERS' COMMENTS:

Reviewer #1 (Remarks to the Author):

The authors have improved the manuscript following my comments and suggestions. Also, the points I raised in the previous round of review have been satisfactorily addressed. I recommend this revised manuscript for publication in Nature Communications.

Reviewer #2 (Remarks to the Author):

I believe the authors have adequately addressed my concerns and I now recommend that the paper be published in Nature Communications.

Incidentally, I see now that the authors already noted in the original manuscript that the transmon was limited by Purcell decay; apologies for missing that. I also agree, in contrast to referee 1, that the authors have demonstrated strong coupling and also resonant (and dispersive) coupling between the transmon and spin qubit.

-Dr. Matthew Reed
HRL Labs